# Symmetry broken and rebroken during the ATP hydrolysis cycle of the mitochondrial Hsp90 TRAP1

Daniel Elnatan[1,2†], Miguel Betegon[1,3†], Yanxin Liu[1], Theresa Ramelot[4], Michael A Kennedy[4], David A Agard[1*]

[1]Department of Biochemistry and Biophysics, Howard Hughes Medical Institute, University of California, San Francisco, United States; [2]Tetrad Graduate program, University of California, San Francisco, United States; [3]Biophysics Graduate program, University of California, San Francisco, United States; [4]Department of Chemistry and Biochemistry, Miami University, Oxford, United States

*For correspondence: agard@msg.ucsf.edu

†These authors contributed equally to this work

Competing interests: The authors declare that no competing interests exist.

**Abstract** Hsp90 is a homodimeric ATP-dependent molecular chaperone that remodels its substrate 'client' proteins, facilitating their folding and activating them for biological function. Despite decades of research, the mechanism connecting ATP hydrolysis and chaperone function remains elusive. Particularly puzzling has been the apparent lack of cooperativity in hydrolysis of the ATP in each protomer. A crystal structure of the mitochondrial Hsp90, TRAP1, revealed that the catalytically active state is closed in a highly strained asymmetric conformation. This asymmetry, unobserved in other Hsp90 homologs, is due to buckling of one of the protomers and is most pronounced at the broadly conserved client-binding region. Here, we show that rather than being cooperative or independent, ATP hydrolysis on the two protomers is sequential and deterministic. Moreover, dimer asymmetry sets up differential hydrolysis rates for each protomer, such that the buckled conformation favors ATP hydrolysis. Remarkably, after the first hydrolysis, the dimer undergoes a flip in the asymmetry while remaining in a closed state for the second hydrolysis. From these results, we propose a model where direct coupling of ATP hydrolysis and conformational flipping rearranges client-binding sites, providing a paradigm of how energy from ATP hydrolysis can be used for client remodeling.

## Introduction

Heat-shock protein 90 (Hsp90) is a highly conserved ATP-dependent molecular chaperone. Although originally identified and named as part of the heat-shock response, Hsp90's many important roles in the general stress response, regulation of protein function, disease and evolution are now appreciated. Hsp90 physically interacts with ~10% of the proteome (*Zhao et al., 2005*), highlighting its ability to recognize a broad range of substrate 'client' proteins, having diverse functions, sequences, structures, and sizes. Although Hsp90 can function as a canonical chaperone and promote protein folding by suppressing aggregation (*Krukenberg et al., 2009*; *Wiech et al., 1992*), it is unique in that it also plays an active role in regulating the activities of a large subset of the proteome, including many proteins involved in signal transduction such as kinases and hormone receptors, thus supporting normal cellular functions. This essential function of Hsp90 is intimately tied to its ATPase activity, and mutations that either enhance or suppress this activity compromise cell viability (*Nathan and Lindquist, 1995*; *Panaretou et al., 1998*). Conversely, deregulation of cellular Hsp90 levels helps support uncontrolled growth in many human cancers making Hsp90 an important pharmacological target (*Whitesell and Lindquist, 2005*).

Hsp90 is a V-shaped homodimer, with each protomer consisting of three major domains: a C-terminal dimerization domain (CTD); a middle domain (MD), which has been linked to client binding; and an N-terminal ATPase domain (NTD). The Hsp90 NTD and the upper part of the MD confer membership to the GHKL (DNA Gyrase, Hsp90, Histidine Kinases, MutL mismatch repair protein) family of ATP-powered molecular machines.

In the apo state, Hsp90 is highly dynamic and can sample a range of conformations from a highly extended to a more compact state, potentially accommodating interactions with a diverse set of client structures and sizes (*Krukenberg et al., 2008*). Large conformational changes are also coupled to nucleotide binding: ATP binding stabilizes a closed state where both NTDs are dimerized, forming the catalytically active state, whereas ADP binding favors a transiently formed compact state (*Shiau et al., 2006*). Such structural studies have suggested an ATPase cycle of closure, hydrolysis and reopening.

Most higher eukaryotic cells possess four Hsp90 homologs: two cytosolic isoforms, one of which is constitutively expressed while the other is induced by heat shock and stress (Hsp90$\beta$ and Hsp90$\alpha$, respectively), one localized to the endoplasmic reticulum (Grp94) and one localized to the mitochondria (TRAP1). Despite some sequence divergence and a few insertions/deletions decorating the globular domains, the structures of all Hsp90 homologs remain conserved, suggesting a fundamental mechanism that adapted to different cellular environments. Indeed, a survey of conformational states in bacterial, yeast, and human Hsp90 revealed species-specific tuning of conformational equilibria and ATPase rates (*Southworth and Agard, 2008*). In general, reaching the NTD-dimerized closed state is rate limiting for ATP hydrolysis. In the case of TRAP1, the formation of this closed state is further regulated by a temperature-sensitive kinetic barrier imparted by an extra N-terminal extension (*Partridge et al., 2014*). For eukaryotic cytosolic Hsp90s, binding of co-chaperones can stabilize particular conformational states and modulate ATPase activity (*Eckl et al., 2013*; *Retzlaff et al., 2010*; *Southworth and Agard, 2011*).

Despite the wealth of information regarding Hsp90 dynamics, little is known about how the energy of ATP hydrolysis is coupled to client protein remodeling. The current model focuses attention on the ATP-induced large conformational change going from a wide-open V-shape to the NTD-dimerized closed state, whereas conversion to the compact ADP state would displace the bound client, followed by reopening to reset the chaperone. However, given that these conformational states are roughly isoenergetic and that ATP binding only marginally stabilizes the closed state (*Mickler et al., 2009*; *Southworth and Agard, 2008*), this model does not provide a clear connection between utilization of energy from ATP hydrolysis and client remodeling.

A recent crystal structure of AMPPNP-bound zebrafish TRAP1 (zTRAP1) in a closed state (*Lavery et al., 2014*) revealed a novel conformational asymmetry between the protomers, most pronounced at the MD:CTD interfaces where one has 2.5-fold more buried surface area than the other (400 Å$^2$ vs. 1000 Å$^2$). One of the protomer arms is buckled while the other remains straight, nearly identical to the previously observed conformation in p23-bound yeast Hsp90 (yHsp90) closed state (*Ali et al., 2006*). Notably, the previously determined client-binding site (*Genest et al., 2013*; *Street et al., 2012*) located at the MD:CTD interface is maximally affected by the protomer asymmetry.

Using TRAP1 as a model system for Hsp90, here we investigate how ATP hydrolysis is coupled to its conformational asymmetry and propose a model connecting it to client remodeling. We examine whether ATP hydrolysis in TRAP1 is sequential, what the order of hydrolysis events is, and how the asymmetry is coupled to the nucleotide states along the ATPase cycle. By covalently linking TRAP1 monomers, we created homogeneous populations of obligate heterodimers having one hydrolysis-dead protomer, and determined that TRAP1 must undergo ATP hydrolysis in both protomers to efficiently progress through the cycle. Crystal structures of WT TRAP1 closed with ATP in the absence of Mg$^{2+}$ showed that the buckled protomer hydrolyzes ATP more rapidly than the straight one. Microsecond-long molecular dynamics simulations reveal differences in water dynamics within the nucleotide-binding pocket of each protomer, highlighting distinct environments surrounding the ATP that may establish differences in hydrolysis rates. Pulsed electron paramagnetic resonance (EPR) distance measurements using dipolar electron-electron resonance (DEER) methods revealed that, in solution, a TRAP1 mutant mimicking the hemi-hydrolyzed (ATP/ADP) state adopts a uniform conformation distinct from the stochastic mixture seen in WT. In this heterodimer, the protomer containing ATP is buckled while the protomer containing the ADP-mimicking mutation is straight. Integrating

this data, we propose a revised model of the TRAP1 ATP-driven cycle where the two hydrolysis events are sequential and deterministic, with the buckled protomer being most competent for ATPase hydrolysis, followed by a flip in the MD:CTD asymmetry to position the opposite protomer in the buckled conformation, promoting hydrolysis of the second ATP and allowing TRAP1 to proceed through the cycle.

## Results

### Two ATP hydrolyses are required by TRAP1 to progress through its ATPase cycle

Previous studies on various Hsp90s show that their ATPase activities follow simple, non-cooperative kinetics despite having two ATP-binding sites (*Dollins et al., 2007*; *Frey et al., 2007*; *McLaughlin et al., 2002*; *Richter et al., 2008*). Consistent with these observations, the activity of the wild-type ATPase domain is unaffected by whether its partner carries a mutation that does not bind ATP or one that does not hydrolyze ATP (*Cunningham et al., 2008*; *Richter et al., 2001*). Such heterodimeric studies are possible because Hsp90 dimers dynamically exchange within minutes (*Hessling et al., 2009*; *Richter et al., 2001*). Simply mixing wild-type and mutant proteins creates a mixture of dimer species (wild-type + heterodimer + mutant) at equilibrium. Using this approach, we looked at human TRAP1 (hTRAP1) ATPase activity by mixing wild-type and a point mutant (E115A) that impairs ATP hydrolysis, but not ATP binding (*Panaretou et al., 1998*) at different ratios. In contrast to previous studies, the activities of these TRAP1 mixtures are lower than what is expected if the two ATPase domains were to act independently (*Figure 1A*), suggesting TRAP1 requires that both bound ATPs be hydrolyzed in its ATPase cycle. Since these experiments were done under steady-state conditions, the decreased activity of heterodimers can be due to either an impaired ability to form the closed state or an impaired ability to exit the closed state, thus changing the rate-limiting step.

To distinguish between these models, one can form heterodimers by mixing and use FRET to directly probe the conformation of the dimer (*Hessling et al., 2009*). However, a significant drawback is that the resulting mixture of dimer species complicates further biochemical or especially structural analyses. To allow precise targeting of protomer-specific mutations and produce a homogeneous population of heterodimers without dimer exchange, a covalently bound heterodimeric hTRAP1 was engineered using SpyCatcher/SpyTag (*Zakeri et al., 2012*) appended after the CTD dimerization domain. Similar to our previous studies (*Partridge et al., 2014*), we can efficiently monitor TRAP1 closure by attaching FRET probes to cysteines engineered into the NTD (E140C) and MD (K413C) of a cysteine-free chaperone (*Figure 1B*). Using the covalent heterodimers, the E115A mutation was introduced in one of the protomers. Appropriate functionality was confirmed by observing that only half of the bound ATP was hydrolyzed under single-turnover conditions (*Figure 1C*). Since dimer closure is required to hydrolyze ATP, this rules out the model where the heterodimer cannot form the closed state. Furthermore, it suggests that the impaired steady-state ATPase of the dimer may be due to an inability to exit the closed state. If true, this would lead to a build up of the closed state under multiple-turnover conditions that can be monitored by FRET (high FRET), whereas constant turnover of the wild-type enzyme would maintain a low FRET signal. Indeed, this is just what is observed upon addition of 2 mM ATP/MgCl$_2$ to either wild-type or +/ E115A TRAP1 (*Figure 1D*). The buildup rate of the heterodimer (0.16 min$^{-1}$) is comparable to the steady-state ATP turnover rate of the cysteine-free wild-type (0.19 min$^{-1}$, *Figure 1—figure supplement 2*), indicating that having only one active ATP site does not affect the kinetics of forming the closed state. Since the closure rate of the +/E115A construct is not affected and dimer closure precedes ATP hydrolysis, the reduced steady-state ATPase rate must be due to a slower process that happens after the dimer is closed.

To directly test the ability of the hemi-hydrolyzed dimer to exit the closed state, a FRET experiment was designed to look at the kinetics of dimer opening after ATP hydrolysis starting from a synchronized closed state population. This is possible because TRAP1 accumulates in the closed state upon addition of ATP in absence of Mg$^{2+}$ (*Figure 1E*), whereupon hydrolysis can be initiated by addition of excess MgCl$_2$ (*Partridge et al., 2014*). If hydrolysis of both bound ATPs is required to reset TRAP1 back to an open state, then the hemi-hydrolyzing mutant should have an impaired re-

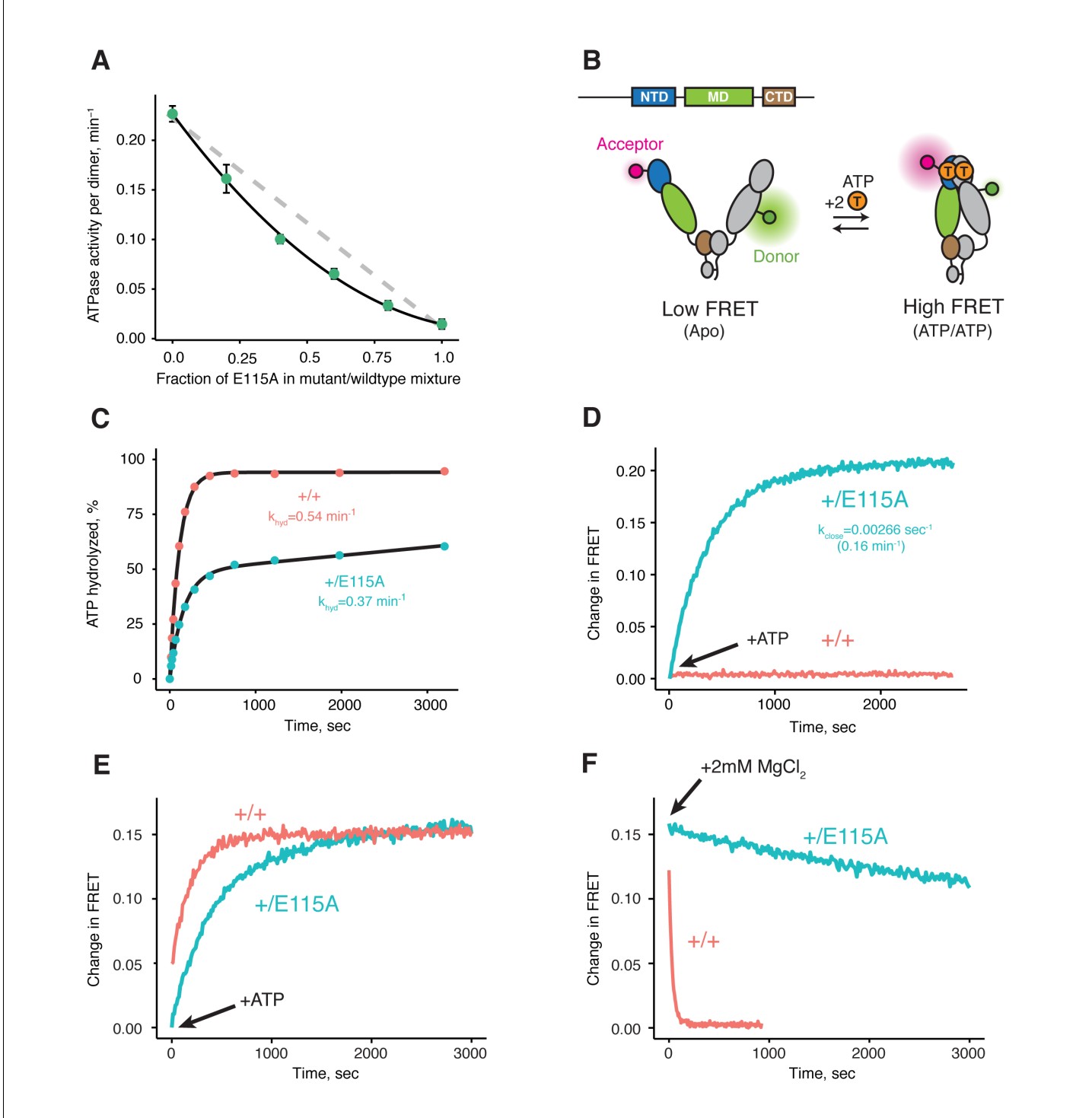

**Figure 1.** Both ATPs need to be hydrolyzed for efficient cycling. (**A**) Steady-state ATPase assay with constant dimer concentration and varying ratios of wild-type and catalytically-dead E115A mutant shows that ATP hydrolysis by each protomer is not independent. Each point and error bar are one standard deviation and averaged from triplicate experiments. Black line is a fit to a binomial distribution of wild-type:mutant:heterodimer, solved for only the heterodimeric activity. The gray dashed line shows the expected activity for independent ATP hydrolysis. (**B**) Covalently linked heterodimers for the FRET assay with fluorescent labels on the NTD and MD (E140C, K413C respectively). (**C**) Single-turnover ATPase kinetics of wild-type (+/+, red) and heterodimeric, hemi-hydrolyzing (+/E115A, blue), show activity of the remaining site is not compromised. Black curves are exponential fits with an additional linear term to account for a slow steady-state activity in the +/E115A. (**D**) FRET assay looking at build up of closed state (high FRET) in the +/ E115A heterodimer (+/E115A, blue) and wild-type (+/+, red) in presence of MgCl$_2$ to allow ATP turnover. The +/E115A data were fit to an exponential

*Figure 1 continued on next page*

*Figure 1 continued*

with the indicated rate, $k_{close}$. (E) FRET assay showing closed state build up in both +/+ (red) and +/E115A (blue) heterodimers after addition of 2 mM ATP in absence of $MgCl_2$. (F) Addition of excess $MgCl_2$ triggers efficient dimer reopening, as measured by FRET, in +/+ (red) but not +/E115A (blue) heterodimers in reactions pre-incubated with ATP without $Mg^{2+}$.

The following figure supplements are available for figure 1:

**Figure supplement 1.** Raw fluorescence intensities of FRET data in *Figure 1D–1F*.

**Figure supplement 2.** Steady-state ATPase assays of cysteine-free hTRAP1 and heterodimeric hTRAP1 (+/+and + /R402A) at 30 ˚C.

opening rate after $MgCl_2$ is added. Indeed, a very slow re-opening rate is observed for the +/E115A heterodimers, whereas the wild-type protein rapidly returns to low FRET baseline (*Figure 1F*). These observations strongly support a model where two ATP hydrolyses are required for TRAP1 to progress efficiently through its ATPase cycle.

## Crystallographic evidence of sequential and deterministic ATP hydrolysis in TRAP1

The observed accumulation of TRAP1 closed state upon addition of ATP in the absence of $Mg^{2+}$ appears to be specific to the mitochondrial homolog, as bacterial Hsp90 (bHsp90) and yHsp90 do not exhibit this behavior and remain in the open state (data not shown). To characterize this unique state and, through it, gain insight into the TRAP1 ATPase cycle, we crystallized zebrafish TRAP1 (zTRAP1) closed with ATP in the absence of $Mg^{2+}$ with an excess of EDTA. An initial crystal, collected 3 days after setting crystal trays, diffracted to 2.5 Å and was solved using molecular replacement with the previously published TRAP1 structure closed with AMPPNP (*Lavery et al., 2014*) (PDB:4IPE). The refined model showed no major conformational differences with any of the previously published full-length TRAP1 structures closed with non-hydrolyzable nucleotides (*Figure 2A*). The same pronounced asymmetry (one straight protomer and one buckled protomer) was clearly observed. However, differences exist in the ATP-binding pocket: as expected, the $Mg^{2+}$ electron density previously observed contacting the nucleotide phosphates is missing and, surprisingly, while the straight protomer contained a mostly intact ATP, the buckled protomer had only weak density for the γ-phosphate (*Figure 2B*, Day 3). This clearly indicated that the buckled protomer preferentially hydrolyzes ATP, but does not inform on whether hydrolysis occurred before or after crystallization.

To differentiate between these possibilities, we took advantage of the fact that our crystals grow overnight, allowing collection and freezing of crystals every 24 hr for a time-resolved experiment. After screening crystals from multiple time points for diffraction quality, two additional datasets were collected (1-day and 53-day-old crystals), solved and refined through the same methods to resolutions of 3.5 Å and 2.2 Å, respectively. Similar to the 3-day-old crystal, no significant conformational differences were observed and no $Mg^{2+}$ ion density was present in the ATP-binding pocket. However, electron densities for the ATP molecules showed a monotonic progression from an ATP-ATP structure (one day old crystal) through an ATP-ADP intermediate (3-day-old crystal) to an ADP-ADP structure (53-day-old crystal) (*Figure 2B*), clearly demonstrating that hydrolysis is occurring in the crystal. This result leads to a model in which TRAP1 hydrolysis events are sequential and deterministic, with the buckled protomer conformation being catalytically favorable and hydrolyzing first. At least in the context of the crystal lattice, no major conformational rearrangements of the symmetry are linked to the first ATP hydrolysis.

## TRAP1 hydrolyzes ATP in solution in absence of magnesium

The fortuitous crystallographic observation of two different ATP hydrolysis rates in the closed state indicates that TRAP1 has the capacity for ATP hydrolysis in the absence of $Mg^{2+}$, albeit slowly. To exclude the possibility that this was somehow unique to the crystal, we needed an extremely sensitive solution assay that could detect TRAP1 ATPase activity in the absence of $Mg^{2+}$ – a level that could be near the rate of spontaneous ATP hydrolysis. A sensitive fluorescent phosphate

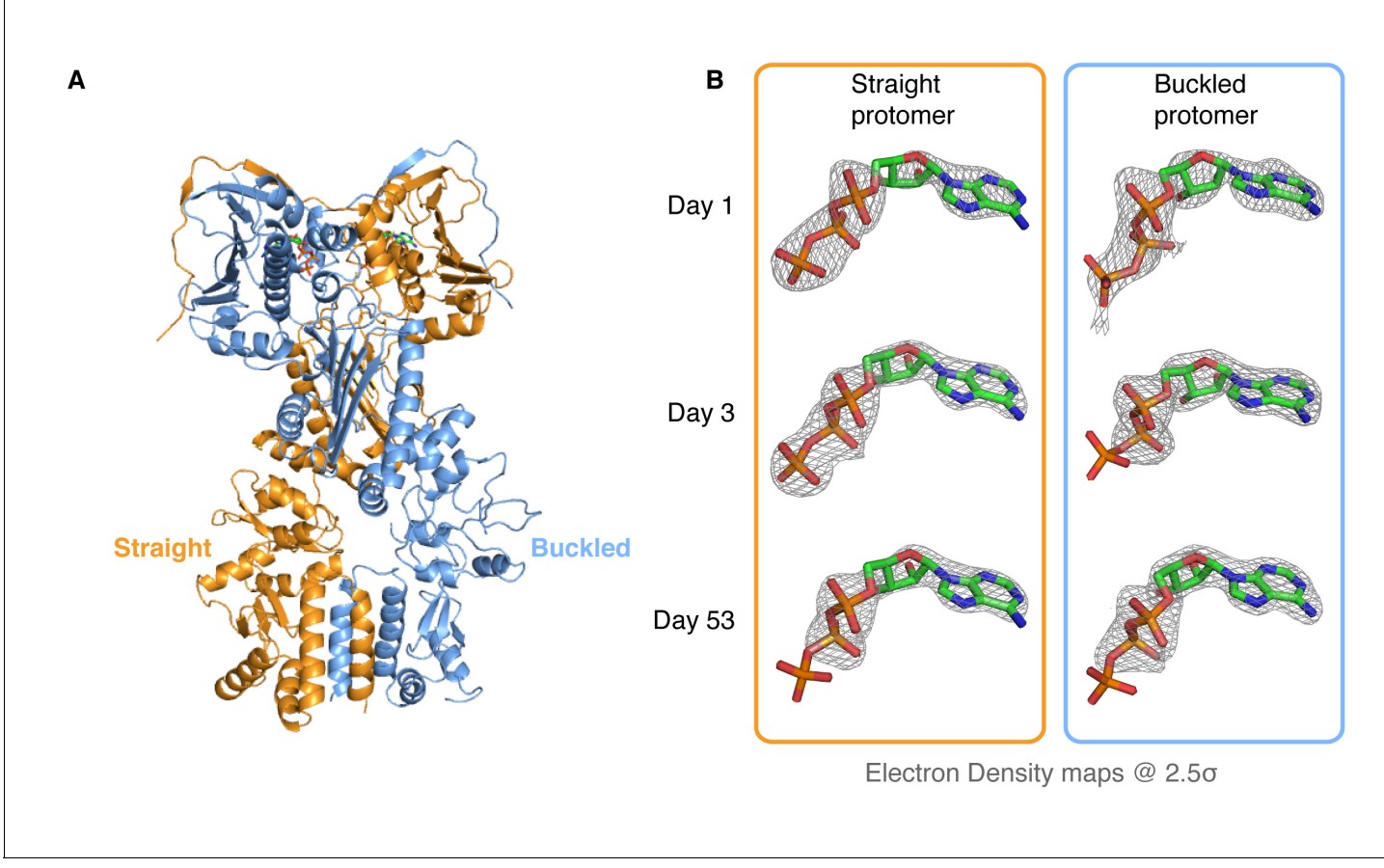

**Figure 2.** Kinetic crystallography indicates that the buckled arm hydrolyzes ATP first. (**A**) 2.3 Å crystal structure of zTRAP1 closed with ATP obtained from a 3-day-old crystal showing minimal conformational changes without $Mg^{2+}$. Buckled protomer is in blue and straight protomer is in orange. (**B**) ATP electron density maps for the buckled and straight protomers from crystals of different ages showing the evolution of in-crystal hydrolysis.

sensor (**Brune et al., 1994**) was chosen for a simple kinetic assay as done by McLaughlin et al. with human Hsp90 (hHsp90). While ideally this would be done using zTRAP1 to match the crystallography, its much weaker $K_m$ would require an order of magnitude higher ATP concentration to trap the closed state than the human homolog (**Figure 3A**). This would be problematic as it introduces a high free-phosphate background from ATP alone. Instead, human TRAP1 was used to detect ATP hydrolysis in solution with 500 µM of ATP in the presence of excess EDTA. At this ATP concentration, there is no significant amount of apo state by size exclusion chromatography (**Figure 3B**) and the ATP titration by FRET estimates at least 80% closed state (**Figure 3A**).

Under this condition, a low but significant ATPase activity is detected in solution (**Figure 3C**). This ATPase is specific to TRAP1, since the rate of hydrolysis scales with TRAP1 concentration (**Figure 3D**), and it can be inhibited by the addition of Radicicol (**Figure 3C**), a potent ATPase inhibitor for Hsp90 (**Leskovar et al., 2008**). From this experiment, the spontaneous and TRAP1-catalyzed ($Mg^{2+}$-free) ATP hydrolysis rates are 0.00155 and 0.5808 per hr (per active site), respectively. Spontaneous hydrolysis would take 62 days, whereas the protein-catalyzed hydrolysis would take 4 hr to hydrolyze 90% of the initial ATP concentration. Although we do not have time points between 1, 3, and 53 days for the crystallography, within an order of magnitude, the solution rates roughly correspond to what we see in the crystal. After 3 days, ATP is fully hydrolyzed by the buckled protomer while ATP remains intact in the straight protomer. Under the same experimental conditions, the rate of dimer closure is an order of magnitude faster (6.95/hr) than the observed hydrolysis rates (**Figure 3—figure supplement 1**). Since dimer closure is no longer the slowest step in hydrolysis, it is

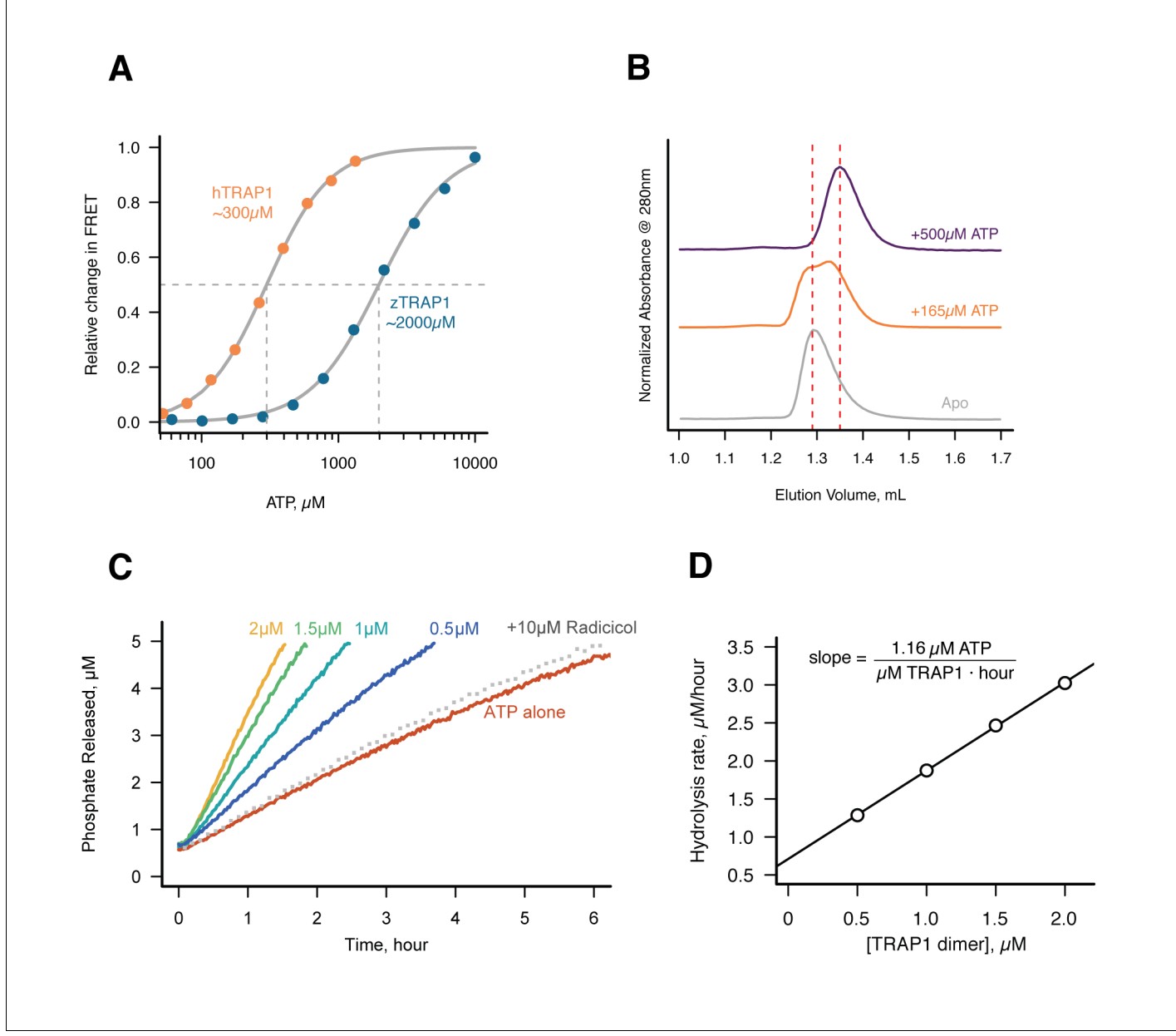

**Figure 3.** Without $Mg^{2+}$ hTRAP1 adopts the closed state and slowly hydrolyzes ATP in solution. (**A**) Equilibrium titration of closure in response to ATP in presence of excess EDTA hTRAP1 (orange) and zTRAP1 (dark blue) using FRET. The indicated half-max concentrations are obtained from fits to the Hill equation (gray lines). (**B**) Size-exclusion chromatography of cysteine-free TRAP1 under apo (gray), and after 1.5 hr incubation at 30°C with 165 μM ATP (partial closure, orange), and 500 μM ATP (full closure, purple). Red dashed lines are guides for apo and closed state peak positions. (**C**) Ultra sensitive assay of ATP hydrolysis using fluorescent phosphate-release assay with PBP-MDCC with ATP alone (red) and varying dimer concentrations of cysteine-free TRAP1, and 2 μM TRAP1 + 10 μM radicicol (gray dotted line). (**D**) Initial rates from phosphate-release kinetics plotted against TRAP1 dimer concentration confirming that the rate above baseline is TRAP1 dependent. The ATPase hydrolysis rate per TRAP1 dimer is 1.16 μM ATP· μM TRAP1$^{-1}$ · hr$^{-1}$.

The following figure supplement is available for figure 3:

**Figure supplement 1.** ATP-induced dimer closure in absence of $Mg^{2+}$ by FRET in human and zebrafish TRAP1.

likely that in absence of $Mg^{2+}$, hydrolysis of the first ATP by the buckled protomer is now the rate-limiting step.

## Asymmetric water dynamics near ATP γ-Phosphate between protomers

To further explore the structural origins of the differential rates of ATP hydrolysis by the two protomers, we performed microsecond all-atom molecular dynamics simulations based on the crystal structure of the asymmetric zTrap1 dimer (PDB ID: 4IYN). We focused on dynamics as no significant differences in the coordinates of the two active sites were observed in the crystal structures. As part of the straight ATP-binding pocket lid was disordered in the crystal structure, the ATP lid regions of both protomers were modeled to be identical and ordered. The ATP analog ADP-AlF$_4$ in the crystal structure was replaced with ATP in the presence of $Mg^{2+}$. The simulations were carried out with explicit water at two different temperatures (310 K and 360 K). The high temperature was used to enhance the conformational sampling.

The structures are intact throughout the simulations at both temperatures and exhibit a high degree of flexibility at the microsecond time scale. To investigate a possible mechanism by which the differential ATP hydrolysis is established, we focused on the environment surrounding the ATP. We monitored the numbers of water molecules within a 5 Å radius of the ATP β- and γ-phosphates along the trajectories, sampled every 3 ns.

On average, the ATP γ-phosphate in the buckled protomer has fewer water molecules in proximity than the one in the straight protomer (*Figure 4A*). At higher temperature, the same trend is maintained despite a larger overlap between the distributions (*Figure 4—figure supplement 1A*). This was unexpected given that both of the nucleotide-binding pockets appear to be essentially identical. The average RMSD between the N-terminal domain of the two protomers is 1.74 Å. The different water dynamics likely arises from differential dynamics of the two protomers.

In addition to the different water occupancy, the most striking difference between protomers observed in the simulations are the water dynamics near the two ATP β- and γ-phosphates. Excluding the $Mg^{2+}$ coordination waters (*Figure 4B,C*), most of the waters spend only a few nanoseconds (*Figure 4B*, green) in the buckled protomer. In contrast, in the straight protomer, there are

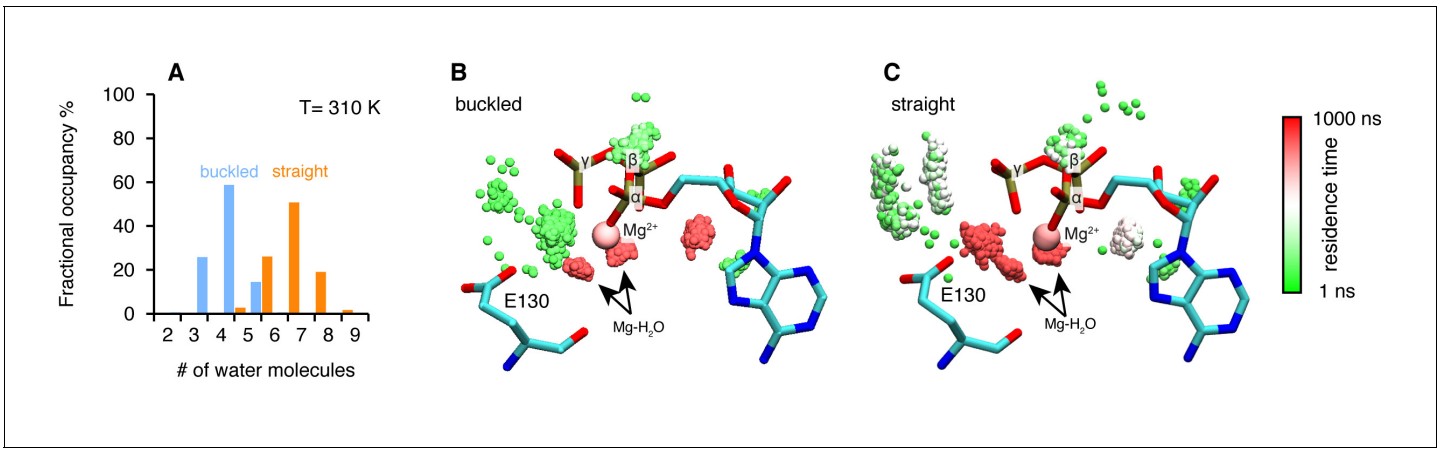

**Figure 4.** Microsecond all-atom molecular dynamics simulations of zTrap1 reveal asymmetric water dynamics near the ATP γ-phosphate. (A) Histogram of water molecules counted near the ATP β- and γ-phosphates (<5 Å) throughout the simulation (each frame is three ns) in the buckled protomer (blue) and the straight protomer (orange) at T = 310 K. (B) and (C) Fractional residence time of each individual water molecule in the ATP-binding pocket for the buckled and straight protomers at T = 310 K, showing significant differences in solvation near the E130. Only the oxygen of water molecules near the ATP β- and γ-phosphates (<5 Å) are shown. Points are accumulated from all frames along the trajectory after aligning the system based on the ATP. Black arrows point to the persistent magnesium-coordinated water molecules, Mg-H$_2$O, which are conserved for the two conformers. Water molecules were colored based on their residence time as indicated by the color bar.
The following figure supplement is available for figure 4:

**Figure supplement 1.** Microsecond all-atom molecular dynamics simulations of zTrap1 reveal asymmetric water dynamics near the ATP γ-phosphate at 360 K.

longer lived waters positioned between E130 and the γ-phosphate (*Figure 4C*, red vs *Figure 4B*, green) and also more waters positioned above E130 (*Figure 4C*, white). This trend is also observed at high temperature (*Figure 4—figure supplement 1B,C*). While the precise mechanism is still unclear, this asymmetry in water occupancy and dynamics between two protomers correlates with the observed preferential hydrolysis of the buckled protomer ATP observed in the time-resolved X-ray crystallography.

## Loss of the γ-Phosphate contact determines MD:CTD conformation in the ATP/ADP state

ATP analogs have been successfully used to capture pre- and post-hydrolysis transition intermediate conformational states of several ATPases providing insights into their mechanisms (*Chen et al., 2007*; *Fisher et al., 1995*; *Wittinghofer, 1997*). In contrast, previous TRAP1 experiments with these analogs resulted in similar asymmetric structures with no obvious conformational differences. In our time-resolved crystallography experiments, the solved structures remain in essentially the same conformation despite having gone through a full conversion from two ATPs to two ADPs. Because crystallographic packing likely prevents any significant conformational changes that might occur after the first ATP hydrolysis, DEER (*Pannier et al., 2011*) was again the optimal choice for directly probing the solution MD:CTD conformation of hTRAP1 in the hemi-hydrolyzed (ATP/ADP) state. The challenge is then to capture a stable ATP/ADP hybrid state.

To generate an ADP-state in only one protomer, we make use of the observation that the MD-Arg (R402 in hTRAP1), the only residue in Hsp90 that contacts the γ-phosphate, acts as an ATP sensor (*Cunningham et al., 2012*) and mutating this residue would effectively mimic the ADP state. Using the SpyCatcher/SpyTag heterodimer, we introduced a point mutant (R402A) to break the γ-phosphate contact in only one protomer. This heterodimeric construct (+/R402A) can form the closed state upon incubation with the ATP analog ADP-BeF (*Figure 5—figure supplement 1*) and has minimally perturbed (twofold increase) ATPase activity (*Figure 1—figure supplement 2*). A cysteine-free version of this heterodimer was used for site-directed spin labeling at the MD (K439C) and CTD (D684C). These positions optimally report on unique MD:CTD distances within each protomer. By measuring the distance between these probes via DEER, one can distinguish between a buckled or straight conformation of the labeled protomer in the closed state (*Lavery et al., 2014*). As expected for the wild-type heterodimer, two peaks were observed centered at about 22 Å and 41 Å with roughly equal proportions (*Figure 5A*). This is close to the 50:50 probability of a protomer randomly adopting either conformation in an asymmetric closed state. Although the observed distances do not exactly match the corresponding values from the crystal structures, they are well within measurement uncertainties when the maleimide linker (~8 Å) and broad distribution of distances are taken into account. Placing this spin-label pair on a protomer carrying the R402A mutation (+/R402A *cis*) or across from that mutant protomer (+/R402A *trans*) can independently report on both MD:CTD conformations of the hemi-hydrolyzed dimer in the closed state.

If TRAP1 adopts a symmetric closed state after losing one γ-phosphate contact, a single distance should be observed irrespective of spin-label placement in the hemi-hydrolyzed state. Instead, two different distances were observed depending on the placement of the spin-labels with respect to the R402A mutation. Protomers with an intact γ-phosphate contact adopt mostly a 22 Å MD:CTD distance - consistent with the buckled conformation (*Figure 5B*), whereas spin-labeled protomers lacking the γ-phosphate contact exhibit a major peak at 41 Å - consistent with the straight conformation (*Figure 5C*). Our interpretation that these distances faithfully report on the MD:CTD conformations rely on assumptions that the spin-labels, the R417A mutation, or the SpyCatcher fusion did not introduce unintended perturbations.

To address this concern, we crystallized an equivalent heterodimeric construct of zTRAP1 (+/R417A in zebrafish sequence numbering) in the presence of ADP-BeF. The heterodimeric crystal diffracted to 3.2 Å and was solved with molecular replacement using the published model of ADP-BeF-bound zTRAP1 (*Lavery et al., 2014*) (PDB:4J0B) and a SpyCatcher-SpyTag complex (*Li et al., 2014*) (PDB:4MLS). As predicted from the DEER results, the heterodimer adopts an essentially identical asymmetric closed state (*Figure 5D*), and the C-terminal-fused SpyCatcher-SpyTag complex packs against its symmetry mate in the opposing dimer (*Figure 5—figure supplement 2*). The structure shows that the R417A mutation does not perturb the overall closed state. To avoid model bias, the dataset was refined starting with alanines replacing arginines at positions 417 on both protomers.

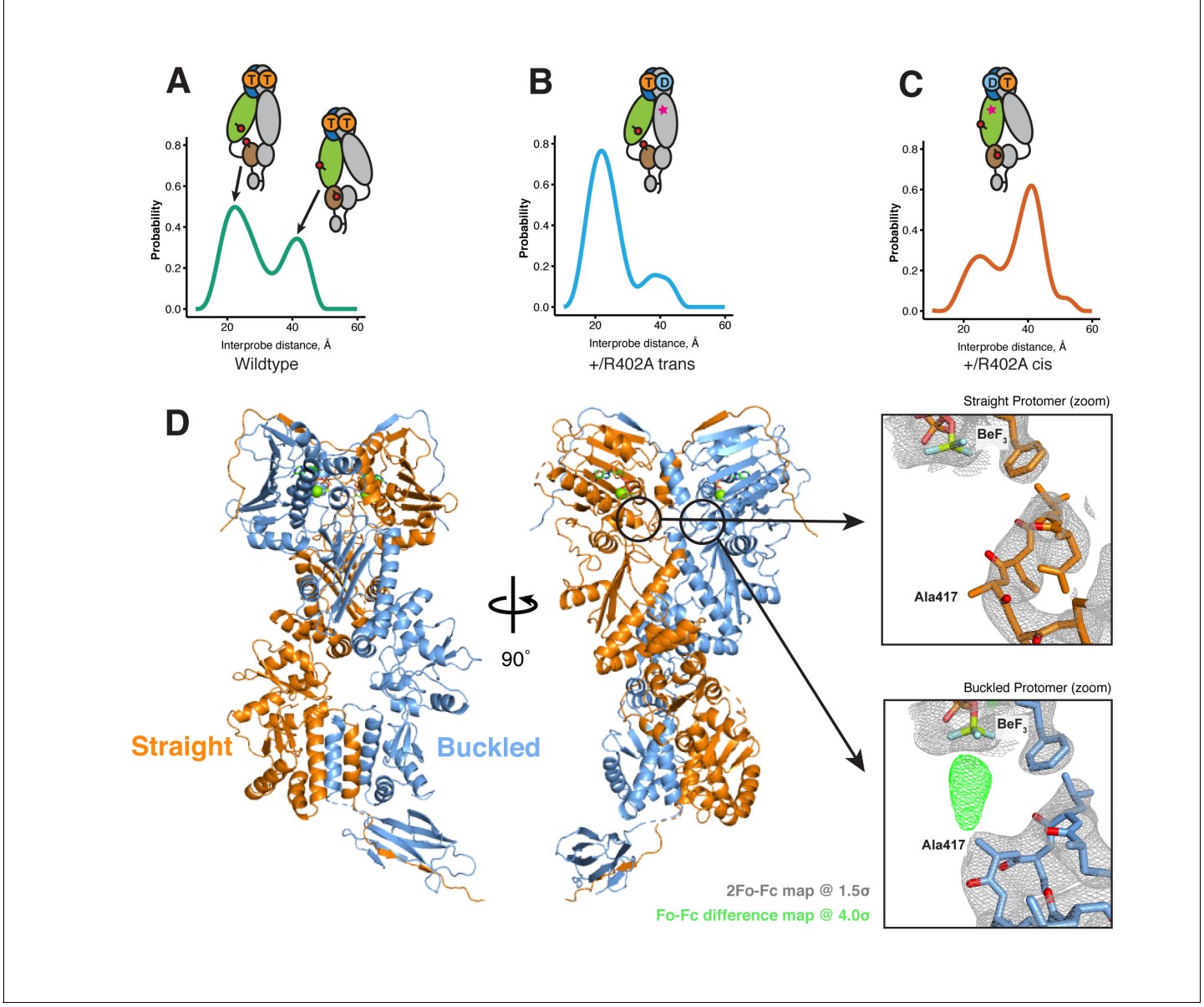

**Figure 5.** The asymmetry is flipped in the ATP/ADP state as revealed by DEER on hemi-hydrolyzed (ATP/ADP) heterodimers. The cartoons depict hTRAP1 heterodimers (one protomer colored by domains and the other in gray; NTD, blue; MD, green; CTD, brown) with spin-labels (red circles) and the relevant nucleotide state (D or T, for ADP or ATP, respectively) as well as the R402A mutation (ADP state mimic; magenta star). The SpyCatcher-SpyTag is shown attached to the CTD tails. (A) +/+heterodimers (green line) partition roughly equally between the buckled (left, 22 Å) and straight (right, 41 Å) conformations. (B) Spin-labels on the opposite (*trans*) protomer of +/R402A heterodimers (blue line) show that nearly all molecules are buckled on the ATP arm. (C) +/R402A heterodimers (orange line) carrying spin-labels on the same (*cis*) protomer as the R402A mutation, showing that the protomer prefers the straight conformation. (D) Crystal structure of +/R417A heterodimeric zTrap1 with the SpyCatcher-SpyTag fusion in the asymmetric closed state showing buckled (blue) and straight (orange) protomers. The dashed line indicates disordered residues. Insets show that the γ-phosphate-sensing R417 is only present on the buckled arm. Phases come from a model having Ala on both protomers. 2Fo-Fc density (gray mesh) and Fo-Fc difference map (green mesh) around the position of the asymmetric R417A mutation. Strong positive density (green mesh) on the difference map is observed only at the buckled protomer.

The following figure supplements are available for figure 5:

**Figure supplement 1.** Interatomic distance distribution, P(r), from SAXS experiments of heterodimeric +/R402A human TRAP1.

**Figure supplement 2.** Crystal packing interactions for the heterodimeric (+/R417A) zTRAP1 fused to the SpyCatcher-Tag domains.

The Fo-Fc difference map (*Figure 5D*, insets) show strong positive density for the arginine only on the buckled protomer (ATP state), while the R417A mutation is on the straight protomer (ADP state). This is completely consistent with the interprobe distances measured in the DEER experiments correctly representing the buckled and straight protomer conformations. Altogether, this demonstrates that the loss of the γ-phosphate contact within the closed dimer is a strong determinant of the corresponding protomer conformation. Thus, the first ATP hydrolysis directly alters dimer asymmetry upon phosphate release.

## Discussion

Previous equilibrium Hsp90 structural studies have provided a picture of a dynamic molecular machine whose conformational ensemble, well described by rigid-body motions of its globular domains, is differentially tuned, but not exclusively determined by nucleotide binding. While ATP-binding has a clear role in stabilizing the NTD-dimerized closed state — depicted clearly by closed states of yHsp90 (*Ali et al., 2006*), zTRAP1 (*Lavery et al., 2014*), and hHsp90 (*Verba et al., 2016*) — the role of ATP hydrolysis in Hsp90's conformational cycle remains unclear. In this study, we present evidence for a sequential, deterministic hydrolysis of the two ATPs within the mitochondrial Hsp90 (TRAP1) dimer. Each step of hydrolysis drives conformational changes at the client-binding site located at the juncture between the middle and C-terminal domains. We discuss below how this provides a new framework for understanding the mechanism of ATP-dependent client remodeling by Hsp90.

### Mechanism of conformational coupling to sequential ATP hydrolysis

Earlier TRAP1 studies had shown how the strain of closing results in a markedly asymmetric (straight: buckled) two-ATP (ATP/ATP) closed state both in the crystal and in solution (*Lavery et al., 2014*). Unlike the stochastic picture that emerged from equilibrium structural studies (*Krukenberg et al., 2008*; *Mickler et al., 2009*; *Southworth and Agard, 2008*), here we show that hydrolysis of both ATPs is required for efficient reopening. Thus, once closed, kinetic rather than thermodynamic processes govern progression through the conformational cycle. Given this observation, we focused on whether the order of ATP hydrolysis depends upon the asymmetry and on what happens after the first ATP is hydrolyzed.

To facilitate this work, covalent heterodimers were efficiently created using C-terminal fusions with SpyCatcher and SpyTag (*Zakeri et al., 2012*). Also critical, was the ability to close TRAP1 with ATP but in the absence of $Mg^{2+}$, allowing the normally rate-limiting closure step to be bypassed, thereby synchronizing all molecules in a closed ATP/ATP state. The fortuitous ability of the $Mg^{2+}$-free closed state to be crystallized with ATP allowed the slow hydrolysis process to be examined crystallographically using crystals of different ages, revealing that the buckled protomer hydrolyzes ATP first while the straight protomer is still bound to an intact ATP. Concerned that the slow hydrolysis seen in the crystal could be an artifact of packing effects on the ATP lid, quantitative kinetics using a highly sensitive assay for phosphate release confirmed that in the absence of $Mg^{2+}$, ATP hydrolysis does occur slowly in solution.

What happens structurally after the first ATP is hydrolyzed? Unfortunately, beyond small perturbations, the crystal lattice blocks significant conformational rearrangements, even after weeks when both ATPs have been hydrolyzed. To resolve this, we used DEER measurements on heterodimers engineered to mimic the mixed ADP/ATP state in solution. In the ATP/ATP state, the labeled protomers adopt roughly an equal mixture of the two conformers, due to the random buckling on transition to the highly strained closed state. By contrast, after the first ATP hydrolysis, essentially all the molecules are in a defined state with the ADP protomer being straight and the ATP protomer buckled.

Combined, these results indicate that hydrolysis of the first ATP must alter the conformational asymmetry in all molecules such that the ADP containing protomer straightens while the remaining ATP protomer adopts the buckled, catalytically favorable conformation. Including our observation that reopening only efficiently occurs from the ADP/ADP state leads to the proposed conformational cycle, revealing how ATP hydrolysis can potentially perform work to remodel a bound client protein (*Figure 6*): (i) Binding of ATP leads to stabilization of a strained, high-energy closed state (*Lavery et al., 2014*; *Partridge et al., 2014*). (ii) The buckled protomer preferentially

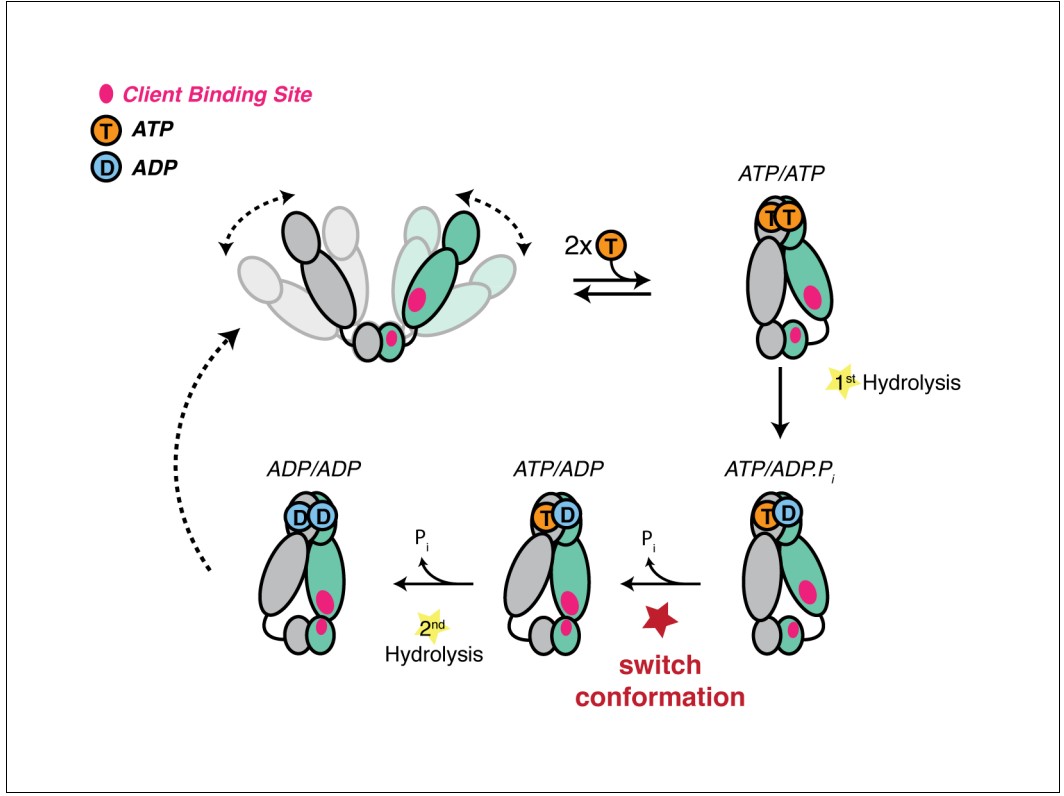

**Figure 6.** Revised model of the TRAP1 ATPase cycle showing the obligatory sequential hydrolysis and conformational switching. The protomers are colored teal and gray. The dynamic apo state (upper left) binds two ATPs, which stabilize a strained asymmetric NTD-dimerized closed state. Within this closed state, ATP is hydrolyzed first by the buckled protomer. Release of Pi likely drives the observed conformational switch of the straight protomer (ATP) to a buckled conformation, while the previously buckled protomer (now ADP), straightens. Concomitant with the flip, the client-binding sites (magenta ellipses) are rearranged, to facilitate client remodeling. Now in a buckled conformation, the second ATP is set up to be hydrolyzed. Finally, the ADP/ADP dimer re-opens, releasing nucleotides and resetting TRAP1 to the apo state.

hydrolyzes ATP, transiently producing a buckled ADP state. (iii) The asymmetry flips so that the ATP protomer is now buckled, setting it up for hydrolysis. (iv) The second ATP is hydrolyzed, leading to reopening and fully relieving the strain. While this work was done with the mitochondrial Hsp90 ortholog, the proposed sequential hydrolysis mechanism also explains the lack of ATPase cooperativity observed in other Hsp90s (*Frey et al., 2007*; *Richter et al., 2008*).

## A model for Hsp90 client maturation mechanism

Although Hsp90 has been studied for over 30 years and its importance in many key cellular processes established, the mechanism by which it matures a large set of diverse clients remains uncertain. This is partly due to Hsp90 preferentially interacting with intrinsically unstable proteins, which has hampered in vitro studies of the chaperone in the context of these clients. In the absence of client proteins, Hsp90 cycles between a wide-open V-shaped apo state and a more compact closed state. In principle, rearrangements of hydrophobic residues accompanying the large conformational change (open-to-closed) could be used to remodel client protein, leading to a molecular 'clamp' model (*Prodromou et al., 2000*). However, we note that constricting the large open-to-close transition, as done by forcing constitutive dimerization of the NTDs, has no major adverse effect on cell growth and only modestly impairs client activation (*Pullen and Bolon, 2011*). This observation, plus the marginal impact of ATP binding in driving dimer closure, the typically quite weak ATP affinity, and the fact that the open and closed states are roughly isoenergetic, implies that the open-to-close transition is unlikely to be the ATP-driven mechanism for client remodeling by Hsp90.

The conformational changes directly linked to ATP-hydrolysis, especially at client-interacting residues, seem better suited for such a mechanism. Studies on HtpG with the model client *Δ131Δ* (*Street et al., 2011*) as well as the recently determined atomic structure of the human Hsp90:Cdc37:Cdk4 kinase complex (*Verba et al., 2016*), have begun to map critical determinants of client-binding residues at the juncture of the Middle and C-terminal domains. As noted by Lavery et al., this coincides with the region of maximal asymmetry in TRAP1 (*Lavery et al., 2014*). Our observation of a hydrolysis-driven conformational switch in which ATP hydrolysis directly promotes a rearrangement of the client-binding region provides a mechanism for actively inducing conformational changes in the client. Equally important is that flipping of the entire TRAP1 population allows clients preferentially bound to either protomer, depending on their size, conformation or other characteristics, to be remodeled via rearrangement of their binding interface. Thus, the first hydrolysis would be used for client remodeling, while the second one would be used to reset the ATP-dependent cycle of TRAP1. Notably, there is some evidence that such a sequential ATP hydrolysis mechanism may be shared among other GHKL ATPases. Yeast topoisomerase II, an enzyme which unlinks double-stranded DNA catenanes, hydrolyzes two ATPs sequentially (*Harkins et al., 1998*), and the first ATP hydrolysis appears to be used to facilitate DNA transport in decatenation (*Baird et al., 2001*). Whether this asymmetric mechanism is conserved throughout all GHKL family members remains to be seen.

A critical feature of our model is the presence of functional asymmetry within the Hsp90 homodimer. Such asymmetry can either be intrinsic to Hsp90 or induced by interactions with other components. Indeed, emerging evidence suggests that functional complexes are preferentially asymmetric (*Ebong et al., 2011*; *Kirschke et al., 2014*; *Li et al., 2011*; *Verba et al., 2016*). For the eukaryotic cytosolic enzymes, this asymmetry can be provided by the numerous Hsp90-specific cochaperones or asymmetric post-translational modifications (*Mollapour et al., 2014*). By contrast, TRAP1 and its bacterial homologs have no known cochaperones. The results and data presented here suggest that, in the case of the mitochondrial and bacterial chaperones, this necessary functional asymmetry may be an intrinsic property encoded by the structural asymmetry of the closed state.

## Materials and methods

### Cloning and protein purification

The constructs for Human and Zebrafish TRAP1 were the same as previously used (*Lavery et al., 2014*). Covalent heterodimeric constructs were made by fusing SpyCatcher and SpyTag (*Zakeri et al., 2012*) domain via an 8-a.a Gly-Ser (GGSGSGSG) linker at the C-terminus of TRAP1. The SpyCatcher domain was obtained from the pDEST14 plasmid obtained via AddGene. The construct used for heterodimeric zebrafish TRAP1 crystallography (+/R417A) has modified linker lengths appended to its C-terminus to account for the offset in starting amino acid residues between SpyCatcher and SpyTag. The SpyCatcher fusion has only a 2 a.a. (GS) linker and the SpyTag fusion has a 6 a.a. (GGSGSS) linker. Proteins were expressed in *E. coli* BL21(DE3)-RIL. Cells were grown in TB media at 37°C to $OD_{600}$ of ~0.8 and then induced with 0.5 mM IPTG for 8–12 hr at 16 °C. Proteins were purified by Nickel-affinity chromatography, anion exchange (MonoQ 10/100 GL, GE, Pittsburgh, PA) and gel filtration (S200 16/60, GE).

Heterodimeric constructs were expressed separately and then combined in roughly stoichiometric amounts after imidazole elution and incubated overnight at 4°C while dialyzing into a low-salt buffer (10 mM Tris pH 8.0, 1 mM DTT). The heterodimers are separated via anion exchange with KCl gradient from 5 mM to 250 mM in 14 column volumes. Heterodimeric peaks were pooled and then incubated with TEV protease overnight at 4°C to cleave the N-terminal His-tag. Finally, proteins were further purified by gel filtration before they were snap frozen with liquid $N_2$.

### Crystallization and data collection

Wild-type zTRAP1 protein at 5 mg/mL in 50 mM HEPES pH 7.5, 50 mM KCl and 1 mM TCEP was incubated with 10 mM ATP and 10 mM EDTA for 1 hr at RT to allow TRAP1 closure before setting 2 μL hanging drops by mixing 1:1 with crystallization condition consisting of 0.2 M Na/K tartrate, 19% (v/v) PEG3350 and 36 mM hexamine cobalt. Heterodimeric (+/R417A) zTRAP1 was incubated with

10 mM ADP-BeF/MgCl$_2$ for 1 hr at 30°C to allow dimer closure. After incubation, samples were spun at 16,000xg for 10 min and transferred to new tubes. The crystals were grown in 0.19 M potassium acetate with varying PEG3350 (20–22%) and benzamidine hydrochloride (5–25 mM) as an additive. For each well condition, three protein concentrations: 0.5, 1, and 1.5 mg/mL were used to set 2 µl (1:1 protein to condition) hanging drops per concentration. All diffraction data were collected at beamline 8.3.1 at the Advanced Light Source in Berkeley, CA (at 1.116 Å wavelength and 91.4K).

## Structure determination and refinement

Wild-type zTRAP1 datasets were indexed using iMosflm 7.2 (*Battye et al., 2011*), and solved using molecular replacement (Phenix 1.10 - Phaser-MR) with PDB:4IPE and refined with phenix. refine (*Adams et al., 2010*) and manual refinement in Coot 0.8.3. Ramachandran statistics for the refined zTRAP1 1-day structure are 94.6% favored, 5% allowed, 0.3% outlier. Ramachandran statistics for the refined zTRAP1 3-day structure are 95.4% favored, 3.5% allowed, 1.1% outlier. Ramachandran statistics for the refined zTRAP1 53-day structure are 94.8% favored, 4.3% allowed, 0.9% outlier.

Heterodimeric (+/R417A) zTRAP1 dataset was indexed using XDS (*Kabsch, 2010*) and initially solved by molecular replacement with PDB:4J0B using using Phaser (simple interface) via CCP4i2-alpha interface (*Winn et al., 2011*). Then a sequential search with the SpyCatcher-SpyTag complex (PDB:4MLS [*Li et al., 2014*]) is performed while keeping the previous solution fixed. Refinements of the heterodimeric zTRAP1 were done with Refmac5 (*Murshudov et al., 1997*) using TLS groups and jelly-body restraints. Ramachandran statistics for the refined zTRAP1 +/R417A structure are 94% favored, 4.2% allowed, 1.7% outlier. Data and refinement statistics for all crystals are summarized in *Supplementary file 1A*.

## FRET sample preparation and experiments

Two cysteines were introduced to the cysteine-free heterodimeric hTRAP1 constructs: one at the NTD at Glu140 (E140C) and another on the MD at Lys413 (K413C). Analogous constructs for the zebrafish TRAP1 was created by introducing point mutants G151C and K428C. Purified proteins were labeled with an equal mixture of Alexa Fluor 555 and Alexa Fluor 647 maleimide (ThermoFisher, Waltham, MA) at 2X molar excess to cysteines and incubated overnight at 4°C. Unreacted dyes were then quenched with addition of 5 mM $\beta$-MeOH and removed using HiTrap desalting column (2 × 5 mL). For FRET experiments, 0.5 µM of labeled heterodimers were used. Measurements were obtained using a Horiba Jobin Yvon FluoroMax four spectrofluorometer equipped with a chilling/heating water bath. Samples were excited at 532 nm and emission wavelengths were collected at 567 nm and 668 nm for donor and acceptor fluorescence, respectively. Relative FRET efficiencies are calculated by taking the ratio of acceptor to donor fluorescence intensity. The change in FRET is the change of this ratio relative to timepoint 0.

## SAXS experiments and data processing

Protein samples were buffer exchanged via size-exclusion chromatography using S200 10/300 GL (GE Healthcare Life Sciences) prior to the experiment. 6 mg/mL of heterodimeric hTRAP1 (+/R402A) was incubated at 30°C for 3 hr with 1 mM ADP-BeF in reaction buffer (20 mM potassium phosphate pH7.0, 50 mM KCl, 2 mM MgCl2, 1 mM DTT). After incubation, samples were spun down at 16,000xg for 10 min and transferred to a new tube. Each experiment was collected with a total of 90-min exposure (15 s x 360 frames) using an in-house (Anton Paar SAXSESS mc$^2$, Graz, Austria) SAXS instrument. Fitting of scattering intensity was done using a custom-written software written in Python2.7 and Fortran (*Elnatan, 2017*; a copy is archived at https://github.com/elifesciences-publications/UCSFsaxs). The software applies smearing correction accounting for slit-collimated geometry of the X-ray beam, and it uses a Bayesian algorithm to choose an optimal D$_{max}$ and smoothness of the P(r) (*Hansen, 2000*). The software also estimates protein molecular mass from an invariant, Q$_r$ (*Rambo and Tainer, 2013*).

## DEER sample preparation and measurements

Cysteine-free variants of heterodimeric hTrap1 were used for site-directed spin-labeling with maleimide spin-labels (4-maleimido-TEMPO, Sigma-Aldrich, Saint Louis, MO) after introduction of

cysteines at positions K439 and D684. Prior to labeling, proteins were incubated with 5 mM DTT for 15 min at 4°C, and DTT was then removed using a HiTrap desalting column (2 × 5 mL) equilibrated with N$_2$-purged labeling buffer (20 mM HEPES pH7.5, 100 mM KCl). Spin labels were incorporated by immediate addition of threefold molar excess spin-label and incubated at 4 °C overnight. Unreacted probes were removed using HiTrap Desalting (2 × 5 mL) columns equilibrated with DEER reaction buffer (20 mM potassium phosphate pH 7.0, 50 mM KCl, 2 mM MgCl$_2$). Separation of free-labels and extent of labeling was assessed by continuous wave (CW) EPR using a Bruker EMX EPR spectrometer (9.83 GHz). Labeled proteins were then buffer exchanged via a 30 kDa MWCO concentrator into the same buffer made in D$_2$O. Spin-labeled hTrap1 (~140 µM dimer) was incubated with 1 mM ADP +2 mM BeF mix (2 mM BeCl$_2$ +10 mM KF) in presence of 2 mM MgCl$_2$ for 1 hr at 30°C. Glycerol-$d_8$ (Sigma-Aldrich) was then added to a final ~30% (v/v) before snap freezing 10 µl samples in 1.1 mm ID quartz capillary tubes in liquid N$_2$. Four-pulse DEER data collection and analysis was done as previously described (*Lavery et al., 2014*). Data analysis was done with DeerAnalysis 2013 (http://www.epr.ethz.ch/software.html) in MATLAB to determine distance distributions as previously described (*Lavery et al., 2014*).

### Production of phoshate-binding protein and labeling for phosphate release assay

Phosphate release was assayed using phosphate-binding protein (PBP) labeled with MDCC (abcam, ab145370, Cambridge, MA) according to Brune *et al.* with modifications in the protein purification and labeling. The PhoS gene was cloned from BL21 E. coli genomic DNA, and DeoD (*E. coli* purine nucleoside phosphorylase, ecPNPase) and DeoB (*E. coli* phosphodeoxyribomutase, ecPDRM) were synthesized using the BioXP3200 platform (SGI DNA, La Jolla, CA) and cloned into pet28a expression vectors with an N-terminal 6xHis tag. Ala197Cys point mutation in PhoS was introduced for site-specific labeling. All the purification and labeling used plastic containers instead of glass to minimize phosphate contamination, which reduces labeling efficiency. Proteins were purified via Ni$^{2+}$-affinity chromatography, and eluted with 400 mM Imidazole, and then run through HiTrap Desalting column equilibrated with 20 mM HEPES pH 7.5, 50 mM KCl. Proteins were then flash-frozen in liquid N$_2$. For labeling, PhoS.A197C (at ~200 µM) was incubated with 1 µM PNPase, 0.5 µM PDRM, 50 µM MnCl$_2$, 50 µM α-D-glucose-1,6-bisphosphate and 0.5 mM 7-methylguanosine (Sigma-Aldrich) to mop-up contaminating phosphate. MDCC dissolved in DMSO was then added stepwise and gently mixed by inverting the tube, totaling up to 2X-molar excess of cysteines. The labeling reaction takes place at room temperature in the dark for 1 hr. Excess dyes were removed by loading the labeling reaction onto a HisTrap (5 mL) column and protein was eluted with a gradient of 20 mM to 500 mM imidazole. Labeled proteins were dialyzed into 20 mM HEPES pH 7.5, 50 mM KCl and concentrated up to ~300 µM before freezing in aliquots.

For the phosphate release assay, a standard curve of free phosphate was prepared with 10 µM PBP-MDCC and varying concentrations of potassium phosphate. Fluorescence measurements were taken in the SpectraMax M5 platereader with SoftMax Pro software for data acquisition. Samples were excited at 375 nm and emission was recorded at 467 nm with 'Low' sensitivity setting. Only fluorescence counts below the detector linearity was used for analysis (<20000 RFU).

### Steady-state ATPase assay and analysis

Steady-state ATPase activity was measured using enzyme-coupled NADH absorbance assay with 1 mM phosphoenol-pyruvate (PEP), 0.18 mM NADH, 30 U/mL of both pyruvate kinase and lactate dehydrogenase. Kinetic absorbance measurements were carried out in Molecular Devices SpectraMax M5, at wavelengths 340 nm for NADH and 420 nm for background. Slopes were obtained from linear fits within linear regimes of each trace. Each reaction volume is 70 µL and a pathlength calibration is applied to convert absorbance to molar concentrations. For heterodimeric mixing experiment with TRAP1, wild-type and mutant proteins were mixed with different ratios while keeping total protein concentration constant. Each mixture was incubated at 30°C for 1.5 hr to allow for dimer exchange. The final reaction has 2 µM dimer in ATPase reaction buffer (20 mM HEPES pH 7.5, 100 mM KCl, 5 mM MgCl$_2$). The total ATPase activity is fitted according to a binomial distribution of dimer species:

$$V_{total} = V_{wildtype} \cdot f_{wildtype}^2 + V_{mutant} \cdot f_{mutant}^2 + V_{heterodimer} \cdot f_{mutant} \cdot f_{wildtype}$$

where V denotes ATPase activity, f denotes the corresponding fractional population of wild-type and mutant. Since protein concentration is constant in all experiments, $f_{wildtype} = 1 - f_{mutant}$. Heterodimeric protein activity is simply the average of $V_{wildtype}$ and $V_{mutant}$ if assuming independence between activities.

## Single-turnover radioactive ATPase assay

For radioactive atpase assays, reactions are initiated with addition of trace amounts [γ-$^{32}$P]ATP (Perkin Elmer, 10 µCi/µL EasyTide Lead, Waltham, MA) mixed with cold ATP totalling up to ~300 µM. The final radioactivity per 30 µL reaction is 0.1 µCi/µL. The ratio of TRAP1 dimer to ATP concentration is kept at 0.5:1, with slight protein excess at concentrations well above the $K_m$ for ATP to ensure single-turnover condition. Time points were taken by chemically quenching 1.5 µL aliquots with an equal volume of 40 mM Tris pH 8.0, 100 mM EDTA, 2% SDS and 2.5 mg/mL proteinase K. Aliquots (1 µl) of quenched reactions were spotted at 1 cm from the bottom of a PEI Cellulose F TLC plate (Millipore, Billerica, MA), and 6% formic acid, 0.5 M LiCl was used as mobile phase. The radioactive phosphate migrates about ~0.8 of plate length from the origin. Radioactive signal was quantified via exposing the plates to a storage phosphor screen (Amersham, Pittsburgh, PA) for ~1 min, and plate images were scanned with Typhoon FLA 9000. Image quantification was done in ImageJ (Wayne Rasband, NIH. https://imagej.nih.gov/ij/).

## Molecular dynamics simulation

MD simulations were performed in explicit solvent using the TIP3P water model (*Jorgensen et al., 1983*) and the CHARMM22 force field with CMAP corrections for protein and ions (*Mackerell, 2004*; *MacKerell et al., 1998*, *2004*). The initial protein structure was modeling based on the crystal structure of zebrafish Trap1 (PDB ID: 4IYN) (*Lavery et al., 2014*) and subsequently solvated in a cubic water box at 150 mM NaCl salinity, neutralized with extra ions employing VMD (*Humphrey et al., 1996*). All simulations were carried out with periodic boundary conditions in a constant particle number, temperature, and pressure ensemble (NPT). The initial energy minimization and equilibration were carried out on general purpose supercomputers using NAMD 2.10 (*Phillips et al., 2005*). The system to be simulated was first subjected to 10000 steps of conjugate gradient minimization and equilibrated for two ns with harmonic restraints applied to all the heavy atoms of the protein. The simulation was then continued for 10 ns without restraints at a constant pressure of 1 bar using Nosé–Hoover Langevin piston barostat and at a constant temperature (310 K or 360 K) maintained using Langevin dynamics with a damping constant of 1.0 ps$^{-1}$. Multiple time stepping was employed with an integration time step of 2.0 fs, short-range forces being evaluated every time step and long-range electrostatics evaluated every three time steps. Cutoff for short-range nonbonded interactions was 10.0 Å; long-range electrostatics was calculated using the particle-mesh Ewald method (*Darden et al., 1993*). All bonds involving hydrogen in the protein were constrained using RATTLE (*Andersen, 1983*), while the geometries of water molecules were maintained using SETTLE (*Miyamoto and Kollman, 1992*). The resulting equilibrated structure was employed as the initial state for production simulations, carried out on the special purpose supercomputer Anton (*Shaw et al., 2008*, *Shaw et al., 2009*) for ~1.1 µs, where constant temperature (310 K or 360 K) and constant pressure (p=1 bar) were maintained. Multiple time stepping was employed, with an integration time step of 2.0 fs. Short-range forces were evaluated every time step and long-range electrostatics every three time steps. Cutoff for the short-range nonbonded interactions was 9.5 Å; long-range electrostatics was calculated using the k-Gaussian Split Ewald method (*Shan et al., 2005*) with a 64 × 64 × 64 grid. All bonds involving hydrogen atoms were constrained using SHAKE (*Ryckaert et al., 1977*).

## Acknowledgements

We thank members of the Agard Lab for helpful discussions. We especially thank John Bruning for collecting diffraction data on heterodimeric zTRAP1 crystals. We also thank Nariman Naber and Roger Cooke for help with CW EPR measurements. We thank Jarett Wilcoxen at the UC Davis CalEPR center for help with DEER measurements. Support for this work was provided by the NIH Protein Structure Initiative–Biology Grant U01 GM098254 (to DAA), U54-GM094597 to (MAK and GT Montelione), the Howard Hughes Medical Institute (to DAA), and an HHMI Helen Hay Whitney

Foundation Postdoctoral Fellowship (to YL). Molecular dynamics simulation was performed using computational resources from the Extreme Science and Engineering Discovery Environment (XSEDE), which is supported by NSF grant ACI-1053575, and the Anton supercomputer at the Pittsburgh Supercomputing Center (PSC) supported by NIH grant R01GM116961. The Anton supercomputer at PSC was generously made available by DE Shaw Research. We also thank James Holton, George Meigs, and staff at Advanced Light Source (ALS) beamline 8.3.1 for help with data collection. Beamline 8.3.1 at the ALS is operated by the University of California Office of the President, Multicampus Research Programs and Initiatives grant MR-15–328599 and Program for Breakthrough Biomedical Research, which is partially funded by the Sandler Foundation.

## Additional information

### Funding

| Funder | Grant reference number | Author |
|---|---|---|
| Howard Hughes Medical Institute | | Daniel Elnatan<br>Miguel Betegon<br>Yanxin Liu<br>David A Agard |
| Helen Hay Whitney Foundation | | Yanxin Liu |
| National Institutes of Health | U54-GM094597 | Michael A Kennedy |
| National Institutes of Health | U01-GM098254 | David A Agard |

The funders had no role in study design, data collection and interpretation, or the decision to submit the work for publication.

### Author contributions

DE, MB, Conceptualization, Formal analysis, Investigation, Methodology, Writing—original draft, Writing—review and editing; YL, Investigation, Methodology, Writing—original draft, Writing—review and editing; TR, Investigation, Methodology, Writing—review and editing; MAK, Resources, Supervision, Funding acquisition; DAA, Conceptualization, Resources, Supervision, Funding acquisition, Investigation, Methodology, Writing—original draft, Project administration, Writing—review and editing

### Author ORCIDs

Daniel Elnatan, http://orcid.org/0000-0002-8359-0522
Miguel Betegon, http://orcid.org/0000-0001-7625-6190
Yanxin Liu, http://orcid.org/0000-0002-2253-3698
David A Agard, http://orcid.org/0000-0003-3512-695X

## Additional files

### Supplementary files

• Supplementary file 1 Data collection and refinement statistics for zebrafish TRAP1 crystals. Each dataset was collected from a single crystal. Values in parentheses are for highest-resolution shell. DOI: 10.7554/eLife.25235.014

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
