## [Decision Letter]

Thank you for submitting your article "Symmetry broken and rebroken during the ATP hydrolysis cycle of the mitochondrial Hsp90 TRAP1" for consideration by *eLife*. Your article has been favorably evaluated by John Kuriyan (Senior Editor) and three reviewers, one of whom, Andreas Martin (Reviewer #1), is a member of our Board of Reviewing Editors. The following individual involved in review of your submission has agreed to reveal their identity: Art Horwich (Reviewer #2).

The reviewers have discussed the reviews with one another and the Reviewing Editor has drafted this decision to help you prepare a revised submission.

Summary:

In the present study, Elnathan et al. investigate the ATPase mechanism of the mitochondrial Hsp90 TRAP1, how its ATP hydrolysis is coupled to dimer asymmetry, and how hydrolysis-coupled conformational switching of protomers between buckled and straight states may rearrange client protein binding sites for remodeling. Using a combination of X-ray structure determination and kinetic crystallography, MD simulations, and elegant biochemical experiments with wild-type TRAP1 and mutant heterodimers, the authors provide strong evidence for sequential and deterministic ATP hydrolysis in the asymmetric dimer, which may induce a conformational flip to drive the ATPase cycle forward and promote the mechanical remodeling of clients. This study thus answers important, long-standing questions about the coordination of ATP hydrolysis and conformational changes of Hsp90.

All reviewers agreed in their overall positive assessments, sharing enthusiasm about this elegant study and its important novel insights into the ATP-hydrolysis cycle of Hsp90. However, they also agreed about revisions that will be necessary before we can make a final decision about publication. These revisions will, for instance, have to address discrepancies in the presented kinetics for dimer closure and ATP hydrolysis, and the derived model for the rate-limiting step in the ATPase cycle. Furthermore, the study in general lacks required statistical analyses, as there is no information provided about the number of measurement repeats, errors, or statistical significance of the results. The authors should also strongly consider to further experimentally support the main conclusion about conformational switching. It may indeed be possible to directly monitor the switch in protomer conformation upon ATP hydrolysis, using a heterodimer that is already available. The reviewers agreed that such measurements would significantly strengthen the conclusions of this manuscript and should be feasible to be completed within a couple of months.

As outlined below, the reviewers also request additional discussions, clarifications, and the presentation of more detailed data, for instance for the FRET-based measurements.

Critical revisions:

The reviewers have suggested that a set of experiments would be valuable to resolve some important questions. These proposed experiments should be considered seriously, and if not considered feasible then the revised manuscript should provide a careful justification for why that is the case.

These questions concern the proposal that preferential ATP hydrolysis in the buckled subunit of the closed asymmetric TRAP1 dimer leads to a conformational flip, in which the buckled, now ADP-bound subunit straightens and the straight subunit becomes buckled to facilitate the second ATP hydrolysis event. However, the authors were so far unable to directly monitor this conformational switch or show any succession of states. The current model is solely based on results for static, non-switching subunits that show differential hydrolysis activities and preferred buckled vs. straight conformations depending on the nucleotide state. The authors state in the discussion that "kinetic rather than thermodynamic processes govern progression through the conformational cycle", but present mainly thermodynamic two state measurements that may also be explained by a shift in equilibrium. Experimentally monitoring the switch would therefore significantly strengthen the main conclusion and model.

By using spin-labeled heterodimers in DEER measurements, it may in fact be possible to directly detect the conformational switching upon ATP hydrolysis. The authors should consider spin-labeling the wild-type subunit in the wild-type / E115A heterodimer. Assuming an equal probability of WT and E115A subunits adopting the buckled conformation after adding ATP, about 50% of the WT subunits should be buckled with a small interprobe distance. ATP hydrolysis upon Mg addition is expected to induce straightening of those WT subunits, which should be accommodated by a shift to the larger interprobe distance. This state would then be stable due to the lack of ATP hydrolysis in the E115A subunit (as shown for the high-FRET closed state in Figure 1). Heterodimers in which the E115A subunit was buckled and the labeled WT subunit was straight after ATP addition should not lead to an inverse change in DEER signal, because the EA subunit is hydrolysis incompetent. Actually showing a shift in interprobe distance that follows the kinetics of ATP hydrolysis in the WT subunit would thus nicely confirm the prosed model.

These DEER experiments on subunit flipping could be performed before and after Mg addition to monitor the start and end state only, or by taking aliquots during the transition to actually detect flipping kinetics, with the prerequisite that snap-freezing of the sample slows down the ATP hydrolysis and conformational changes enough to preserve the not yet flipped population during measurement.

Other important revisions:

The reviewers have raised a number of other issues that we feel could be addressed through careful revision of the manuscript, with explanations provided as needed, but without necessarily engaging in additional experiments.

1) Errors and number of repeats should be reported for all measurements.

2) The authors convincingly show by kinetic crystallography that the buckled protomer in the asymmetric TRAP1 dimer hydrolyzes ATP in the absence Mg more rapidly than the straight protomer. However, what is the evidence that these differences in Mg-independent hydrolysis directly translate to the Mg-dependent hydrolysis, which occurs an order of magnitude more rapidly? The authors should comment on potential mechanistic similarities between Mg-independent and Mg-dependent ATP hydrolysis by Hsp90 that would support this assumption. Furthermore, there is some concern about the direct comparison between hTRAP1 and zTRAP, which show a ~ 10-fold difference in Km.

3) The authors performed MD simulations on buckled and straight protomers to explain the differential rates in ATP hydrolysis. However, for those MD simulations the authors replaced the ADP-AlF4 in the crystal structure with ATP-Mg and modeled the ATP lid region of both protomers to be identical and ordered. The authors should at least comment on why it is valid to assume that the nucleotide replacement does not affect the local structure and dynamics of the ATPase sites, and why they assume that fixing the ATP lid, which directly faces the ATP phosphates and is disordered in the straight protomer, is not expected to have a significant effect on the MD simulations (in particular for the straight protomer). It is surprising to me that having more and longer-lived waters in the straight compared to the buckled protomer would lead to slower hydrolysis.

4) The authors present in Figure 1 the single-turnover ATP hydrolysis of the WT / E115A heterodimer (induced by addition of hot/cold ATP-Mg), with a kcat of 0.37 min-1. In Figure 1, they present the kinetics for the buildup of the closed state of WT / E115A, and discuss that "The buildup rate of the heterodimer (0.16 min-1) is comparable to the steady-state ATP turnover rate of the cysteine-free wildtype (0.19 min-1, [Supplementary-material SD1-data]), indicating that having only one active ATP site does not affect the kinetics of forming the closed state."

However, it is unclear why the formation of the closed state for WT /E115A should show the same rate constant as steady-state ATP hydrolysis by WT TRAP1. Due to the hydrolysis-incompetent E115A subunit that traps the closed state (cf. Figure 1), the closed-state formation can be considered single-turnover and should show the same kinetics as single-turnover ATP hydrolysis (0.37 min-1). Both experiments were started by the addition of ATP-Mg, and since forming the closed state precedes hydrolysis, its kinetics should be as fast or faster. The authors should explain the observed 2-fold difference between those rates.

5) Figure 3 shows the kinetics of dimer closure induced by ATP / no Mg, with a rate of 0.116 min-1. Considering that dimer closure precedes hydrolysis, as proposed in the authors' model, why is single-turnover ATP hydrolysis (induced by addition of ATP-Mg) almost 5-fold faster (k = 0.54 min-1, Figure 1)? The dimer closure is also ~ 1.6 fold slower than the steady-state hydrolysis-rate of TRAP1 (0.19 min). Is this difference significant and indicating some effect of Mg on dimer closure, or just experimental error?

Does the difference in rate for dimer closure of WT TRAP1 and the WT / E115A heterodimer upon ATP (no Mg) addition (Figure 1) indicate some additional effect of the EA mutation, besides eliminating ATP hydrolysis?

In the Discussion the authors mention that dimer closure is normally rate-limiting, which could be bypassed by adding ATP in the absence of Mg. But we wonder how the closure step can be rate-limiting, if single-turnover ATP hydrolysis (which should include domain closure) is significantly faster than steady-state hydrolysis (0.54 min-1 vs. 0.19 min-1).

6) When proposing their model, the authors state that "two ATP hydrolyses are required for TRAP1 to progress efficiently through its ATPase cycle". However, this is contradicted by the WT / E402A heterodimer, which shows stimulated ATP hydrolysis ([Supplementary-material SD1-data]), despite the E402A mutation inducing a permanent ADP state and significantly reducing the ATP hydrolysis rate of the protomer / homodimer. This may indicate that both protomers have to adopt the ADP state, but not necessarily hydrolyze ATP, in order to progress efficiently through the ATPase cycle (i.e. one protomer could remain ADP bound throughout the entire cycle).

7) Figure 1: Is the deviation significant? How was the error in the ATPase activity determined? The authors should present statistics on at least 3 repeats of the experiment, which is particularly important given the low ATPase activities. What was die ATP concentration? Is ATP definitely saturated? Could this effect be explained by a shift in KM or vmax?

8) Figure 1: How was the functionality of the labelled proteins tested? Especially as the Cys-light version already has only 20% ATPase.

9) Figure 1: How is the +/E115A trace normalized?

10) Figure 1, Figure 3, and Figure 3—figure supplement 1: What is "Change in FRET"? Are the authors referring to FRET efficiency? Please show the absolute acceptor and donor fluorescence here and also in Figure 1. Otherwise the significance of the changes cannot be judged, and it is important to convincingly rule out quenching etc.

11) Figure 3: How is the 80% closed state calculated? There seems to be more than 20% overlap judged by eye.

12) The R402 in hTRAP1 solely contacts the γ-phosphate, but how is this knowledge enough to state that R402A mimics the ADP state?

13) Why is closing in Figure 5—figure supplement 1 not shown with the FRET-assay (Figure 1), which would in addition give kinetic information?

14) Figure 5: The authors claim that nearly all molecules are buckled on one side in 5B, 5C. Please quantify. This might be even less than 80%, especially in 5C.

---

## [Author Response]

Critical revisions:

*The reviewers have suggested that a set of experiments would be valuable to resolve some important questions. These proposed experiments should be considered seriously, and if not considered feasible then the revised manuscript should provide a careful justification for why that is the case.*

[...]

*These DEER experiments on subunit flipping could be performed before and after Mg addition to monitor the start and end state only, or by taking aliquots during the transition to actually detect flipping kinetics, with the prerequisite that snap-freezing of the sample slows down the ATP hydrolysis and conformational changes enough to preserve the not yet flipped population during measurement.*

We agree with the reviewers that such a direct observation of the conformational switch upon ATP hydrolysis would be an excellent addition. Unfortunately the time scale of the kinetic switching is too fast even at reduced temperatures to permit the necessary handling required for following the switch post adding Mg. We had actually put significant effort into this previously. Since the review, we have now spent many months in an effort to carry out the proposed experiment using the hemi-hydrolyzing heterodimer (+/E115A). Unfortunately, despite repeated efforts and numerous control experiments, the behavior of this mutant in terms of the structural asymmetry as revealed by the DEER profiles did not allow for a clear interpretation of the results. That is, there appears to be an unexpected interaction between the mutation and the site of the spin probes. In an effort to sort out what was happening, we have placed an inactive version of the spin label on the opposite side of the active label on the protomers cis and trans to the E115A mutation and also swapped the SpyCatcher/SpyTag to rule out possible bias in conformation. All the results with the WT and Arg mutant data shown originally were recapitulated in the new experiments. We even went so far as to collect data on a different DEER instrument located in a different lab. To use this mutant for experiments beyond its inability to hydrolyze ATP will require further characterization including choosing new sites for the DEER probes and is unfortunately beyond the scope of this study.

*Other important revisions:*

*The reviewers have raised a number of other issues that we feel could be addressed through careful revision of the manuscript, with explanations provided as needed, but without necessarily engaging in additional experiments.*

*1) Errors and number of repeats should be reported for all measurements.*

We’ve now included the number of repeats for experiments in Figure 1. The FRET experiments shown in the rest of the manuscript are representative single-point experiments. As requested by the reviewers in point #10, we have included the raw intensities to show bona fide FRET phenomena.

*2) The authors convincingly show by kinetic crystallography that the buckled protomer in the asymmetric TRAP1 dimer hydrolyzes ATP in the absence Mg more rapidly than the straight protomer. However, what is the evidence that these differences in Mg-independent hydrolysis directly translate to the Mg-dependent hydrolysis, which occurs an order of magnitude more rapidly? The authors should comment on potential mechanistic similarities between Mg-independent and Mg-dependent ATP hydrolysis by Hsp90 that would support this assumption. Furthermore, there is some concern about the direct comparison between hTRAP1 and zTRAP, which show a ~ 10-fold difference in Km.*

We made this assumption because the crystal structures of the asymmetric closed state with ATP-bound (no magnesium) and AMPPNP-bound with magnesium are identical, with the exception of the positions of water molecules near the phosphate groups of the nucleotide. Since the dimer reaches the same closed state irrespective of magnesium binding to the nucleotide, we can only speculate that the Mg-dependent hydrolysis happens more rapidly due to better water coordination by the metal. As for the concern regarding the use of human and zebrafish TRAP1, the 10-fold difference in apparent nucleotide affinity may just be due to species-specific tuning that exists in Hsp90 family. Using small-angle X-ray scattering, we previously showed that the asymmetric closed state is conserved across species (bacterial Hsp90 vs. human TRAP1) and also within orthologs (human vs. zebrafish TRAP1) (Lavery et al., 2014). Since the focus of this work is to look at the conserved asymmetric closed-state within these two species, we feel that it’s appropriate to use either one as a model system, especially when it is critical to enable the experiments (for example the crystallization).

*3) The authors performed MD simulations on buckled and straight protomers to explain the differential rates in ATP hydrolysis. However, for those MD simulations the authors replaced the ADP-AlF4 in the crystal structure with ATP-Mg and modeled the ATP lid region of both protomers to be identical and ordered. The authors should at least comment on why it is valid to assume that the nucleotide replacement does not affect the local structure and dynamics of the ATPase sites, and why they assume that fixing the ATP lid, which directly faces the ATP phosphates and is disordered in the straight protomer, is not expected to have a significant effect on the MD simulations (in particular for the straight protomer). It is surprising to me that having more and longer-lived waters in the straight compared to the buckled protomer would lead to slower hydrolysis.*

The 1-day zTrap1 crystal structure with ATP bound reported here confirmed that replacement with ATP does not affect the local structure. In fact, all crystal structures of the closed states of TRAP1 obtained with ATP or multiple ATP analogs are essentially identical regarding the lid structures, so it’s safe to assume that nucleotide replacement does not significantly alter the local structures within the closed state. The part of the lid that is disordered and we modeled is not in direct contact with ATP (~15 Å away). Starting with identical lid conformations allows us to remove bias from different initial lid conformations in the simulations. In our MD simulations, the lid in the buckled protomer consistently exhibits more flexibility than the one in the straight protomer despite starting from the same state. This indicates that the simulation is long enough to overcome any bias from the initial structures. The observation that longer-lived waters correlated with slower hydrolysis also surprised us. The difference in water dynamics is only a correlation with hydrolysis rate. The precise mechanism remains to be explored, possibly through in-depth QM/MM studies.

*4) The authors present in Figure 1 the single-turnover ATP hydrolysis of the WT / E115A heterodimer (induced by addition of hot/cold ATP-Mg), with a kcat of 0.37 min-1. In Figure 1, they present the kinetics for the buildup of the closed state of WT / E115A, and discuss that "The buildup rate of the heterodimer (0.16 min-1) is comparable to the steady-state ATP turnover rate of the cysteine-free wildtype (0.19 min-1, [Supplementary-material SD1-data]), indicating that having only one active ATP site does not affect the kinetics of forming the closed state."*

*However, it is unclear why the formation of the closed state for WT /E115A should show the same rate constant as steady-state ATP hydrolysis by WT TRAP1. Due to the hydrolysis-incompetent E115A subunit that traps the closed state (cf. Figure 1), the closed-state formation can be considered single-turnover and should show the same kinetics as single-turnover ATP hydrolysis (0.37 min-1). Both experiments were started by the addition of ATP-Mg, and since forming the closed state precedes hydrolysis, its kinetics should be as fast or faster. The authors should explain the observed 2-fold difference between those rates.*

The experiment done in Figure 1 (single-turnover) was done with a TRAP1 heterodimer construct with native cysteines at high protein concentration (2-orders of magnitude higher than FRET/steady-state ATPase). At high concentration, dimer-dimer interactions between TRAP1 may affect the ATPase rate. In any case, the figure is only meant to show that the E115A mutant works as expected in that it should abolish only one active site. Because the FRET/DEER experiments onward require the use of cysteine-free constructs, the relevant comparison for the FRET experiments is between cysteine-free TRAP1 and the labeled proteins. Since closure precedes hydrolysis, the observation of similar rates between heterodimer (wt/E115A) closure kinetics and steady-state turnover rate of cys-free TRAP1 suggests that indeed closure is rate limiting, since the cys-free TRAP1 is not trapped in the closed state and goes through the cycle.

*5) Figure 3 shows the kinetics of dimer closure induced by ATP / no Mg, with a rate of 0.116 min-1. Considering that dimer closure precedes hydrolysis, as proposed in the authors' model, why is single-turnover ATP hydrolysis (induced by addition of ATP-Mg) almost 5-fold faster (k = 0.54 min-1, Figure 1)? The dimer closure is also ~ 1.6 fold slower than the steady-state hydrolysis-rate of TRAP1 (0.19 min). Is this difference significant and indicating some effect of Mg on dimer closure, or just experimental error?*

*Does the difference in rate for dimer closure of WT TRAP1 and the WT / E115A heterodimer upon ATP (no Mg) addition (Figure 1) indicate some additional effect of the EA mutation, besides eliminating ATP hydrolysis?*

*In the Discussion the authors mention that dimer closure is normally rate-limiting, which could be bypassed by adding ATP in the absence of Mg. But we wonder how the closure step can be rate-limiting, if single-turnover ATP hydrolysis (which should include domain closure) is significantly faster than steady-state hydrolysis (0.54 min-1 vs. 0.19 min-1).*

The main issue of roughly 2-fold difference in ATPase rates vs. closure rate highlights a real difficulty in directly comparing kinetic rates between experiments with different conditions and techniques. We have previously observed that the closure rate of TRAP1 is profoundly sensitive to perturbations in the nucleotide pocket: the rate of closure obtained with AMPPNP, ATP-BeF, ATP no Mg, etc. all differ by as much as an order of magnitude on the same construct (Partridge et al., 2014). The precise mechanism for this effect is still unknown. Subtle effects introduced by mutations in the construct (e.g. Glu115Ala mutation) or experimental conditions (e.g. omitting Mg^2+^) further compound the effect and make comparison across experiments with different constructs impossible.

In Figure 1, we only mean to show that both WT and E115A achieve the same amount of closed state. We show this through a kinetic trace rather than a single end-point to also show that they both start from the same open state and exhibit a similar trajectory. The claim that dimer closure is the rate-limiting step in the ATPase cycle is well-established for Hsp90s and has been previously investigated in TRAP1 by Partridge and Lavery, 2014 (*eLife*).

*6) When proposing their model, the authors state that "two ATP hydrolyses are required for TRAP1 to progress efficiently through its ATPase cycle". However, this is contradicted by the WT / E402A heterodimer, which shows stimulated ATP hydrolysis ([Supplementary-material SD1-data]), despite the E402A mutation inducing a permanent ADP state and significantly reducing the ATP hydrolysis rate of the protomer / homodimer. This may indicate that both protomers have to adopt the ADP state, but not necessarily hydrolyze ATP, in order to progress efficiently through the ATPase cycle (i.e. one protomer could remain ADP bound throughout the entire cycle).*

We don’t see this as a contradiction because in the context of an ATPase cycle in the wildtype protein, the ADP:ADP-state can only be reached from an ATP:ATP-state through 2 hydrolysis events.

*7) Figure 1: Is the deviation significant? How was the error in the ATPase activity determined? The authors should present statistics on at least 3 repeats of the experiment, which is particularly important given the low ATPase activities. What was die ATP concentration? Is ATP definitely saturated? Could this effect be explained by a shift in KM or vmax?*

The data shown in Figure 1 was averaged from three technical replicates. The small error bars shown are one standard deviation from these replicates. ATP concentration was at a saturating amount (2 mM ATP).

*8) Figure 1: How was the functionality of the labelled proteins tested? Especially as the Cys-light version already has only 20% ATPase.*

We don’t know where this 20% comes from as we never claimed this. In fact, the functionality of constructs illustrated in Figure 1 have been checked by both FRET closure and ATPase assays here and are fully functional. The Cysteine-free human TRAP1 and FRET constructs have been characterized in our previous work (Lavery et al., 2014 and Partridge et al., 2014).

*9) Figure 1: How is the +/E115A trace normalized?*

The intensity of migrating spots of ^[31]^P and [γ-^[31]^P]-ATP was quantified per time-point on a TLC lane. The plot shown in Figure 1 for +/E115A curve is the fractional integrated intensity of the phosphate spot over the total intensity per lane. After 3000 seconds, the phosphate spot is ~50% of the total intensity, whereas in the +/+ case almost all of the signal is now at the phosphate spot.

*10) Figure 1, Figure 3, and Figure 3—figure supplement 1: What is "Change in FRET"? Are the authors referring to FRET efficiency? Please show the absolute acceptor and donor fluorescence here and also in Figure 1. Otherwise the significance of the changes cannot be judged, and it is important to convincingly rule out quenching etc.*

We quantify FRET as the ratio of acceptor over donor intensity. The change in FRET is the change of this ratio relative to that of time point zero. We have included this clarification in the Materials and methods section, as well as the raw donor and acceptor fluorescence traces showing anti-correlation, as expected for a FRET phenomenon.

*11) Figure 3: How is the 80% closed state calculated? There seems to be more than 20% overlap judged by eye.*

We apologize for the confusion. Since it’s difficult to judge% closed by chromatography due to significant peak overlap, we meant to refer to the ATP vs. FRET titration in Figure 3. The figure reference in main text has now been fixed.

*12) The R402 in hTRAP1 solely contacts the γ-phosphate, but how is this knowledge enough to state that R402A mimics the ADP state?*

The R402A mutation disables sensing of a γ-phosphate by a protomer (e.g. such that ADP-Pi becomes ADP). In this context, the Arg-to-Ala substitution on the MD effectively mimics the addition of ADP to the wildtype protein: homodimers with this point mutant remain in the open state in presence of ATP, while the WT/R402A heterodimer accumulates in the closed state in the presence of ATP alone and reopens upon addition of Mg – as expected if it were properly mimicking the ADP state. Additionally, as evidenced by the fact that the nucleotide pocket can accommodate an ATP analog with a dye moiety attached at the γ-phosphate (Ratzke et al. 2012, see both references below), there are no steric constraints around the γ-phosphate that could be used to discriminate between nucleotide states. This indicates that the MD-Arg is the only structural determinant that differentiates ATP vs. ADP states.

*13) Why is closing in Figure 5—figure supplement 1 not shown with the FRET-assay (Figure 1), which would in addition give kinetic information?*

While kinetic data with FRET might provide extra information, the purpose of Figure 5—figure supplement 1 is simply to show that the heterodimer can adopt the closed state in solution.

*14) Figure 5: The authors claim that nearly all molecules are buckled on one side in 5B, 5C. Please quantify. This might be even less than 80%, especially in 5C.*

We feel that precise quantitation of the distance distribution is not necessary and also very challenging with DEER. For clarity, we have changed the wording in our Figure legend for 5C from “showing nearly all.. adopt.. buckled on the ATP protomer” to “showing that the protomer prefers the straight conformation.”

References

Ratzke, C., Berkemeier, F. and Hugel, T. Heat shock protein 90's mechanochemical cycle is dominated by thermal fluctuations. PNAS 109, 161–166 (2012).

Ratzke, C., Nguyen, M. N. T., Mayer, M. P. and Hugel, T. From a ratchet mechanism to random fluctuations evolution of Hsp90's mechanochemical cycle. Journal of Molecular Biology 423, 462–471 (2012).